# Willingness to Receive COVID-19 Vaccination in Japan

**DOI:** 10.3390/vaccines9010048

**Published:** 2021-01-14

**Authors:** Takeshi Yoda, Hironobu Katsuyama

**Affiliations:** 1Department of Public Health, Kawasaki Medical School, Kurashiki 701-0192, Japan; katsu@med.kawasaki-m.ac.jp; 2Department of Health and Sports Science, Kawasaki University of Medical Welfare, Kurashiki 701-0193, Japan

**Keywords:** COVID-19, vaccine, willingness, hesitancy, Japan

## Abstract

In the wake of the COVID-19 pandemic, vaccines are being developed by many countries for the safety of their population. However, people of various nations have revealed hesitancy towards being vaccinated, citing reasons such as side effects, safety, a lack of trust in vaccine effectiveness, etc. This study aimed to explore the willingness of people in Japan to be vaccinated or not be vaccinated and the reasons for either decision. A sample of 1100 respondents was drawn from an internet research panel, and a questionnaire survey was administered to evaluate their willingness to be vaccinated by gender, age group, place of living, and underlying illness history. After using descriptive statistics and the chi-squared test to evaluate categorical variables, 65.7% of the participants indicated a willingness to be vaccinated; among them were older age groups, those in rural areas, and those with underlying medical conditions. In addition, males showed less hesitancy towards being vaccinated. Although selectivity bias exists, this study is the first to examine the willingness of Japanese people to be vaccinated. Since vaccine hesitancy and refusal ratio were found to be higher in Japan than in other countries, policy efforts are needed to make the country’s vaccination program viable.

## 1. Introduction

Since the first case of the novel coronavirus infection was reported from Wuhan, China in December 2019 [1], the disease has drastically spread not only in China, but also globally. The World Health Organization called the new virus COVID-19 in February 2020 [2]. At this time, over 42,000 cases had been detected from China, but few cases had been detected from other countries. Toward the end of March, COVID-19 had become a pandemic and spread to over 203 countries and territories, and community transmission had occurred in countries such as the United States, Germany, France, Spain, Japan, Singapore, South Korea, Iran, and Italy [3].

Simultaneously, COVID-19 vaccines are being developed by many countries. At the end of November, two American companies announced that they had developed COVID-19 vaccines with 90–95% effectiveness [4,5]. Other companies in other countries are also currently developing new COVID-19 vaccines [6,7,8]. Although COVID-19 vaccines will soon be available, it is very important to understand whether people are willing to be vaccinated against COVID-19 or not, as this can have large consequences for the success of the vaccination program and potentially large health and economic consequences [9]. To successfully reduce the prevalence and incidence of vaccine-preventable diseases (VPD), vaccination programs rely on a high uptake level. In addition to direct protection for vaccinated individuals, high vaccination coverage rates induce indirect protection for the overall community, or herd immunity, by slowing the transmission of VPD, and thereby decreasing the risk of infection among those who remain susceptible in the community [10]. However, while most people are vaccinated actively, some individuals or groups refuse or hesitate to be vaccinated. A previous study [11] has suggested that the factors of vaccine hesitancy can be classified into three categories: (1) the risk-benefit of vaccines; (2) knowledge and awareness issues; and (3) religious, cultural, gender, or socio-economic factors. Major issues were fear of side effects, distrust in vaccination, and lack of information on immunization or immunization services [12].

The key to successful COVID-19 vaccination is to reduce the vaccine hesitancy ratio. To determine the potential number of people with vaccine hesitancy and their reasons for this sentiment, we conducted an internet-based questionnaire survey in Japan.

## 2. Materials and Methods

This study was conducted in September 2020 in Japan. We used internet research panel data from QiQUMO that is operated by Cross Marketing Inc., Tokyo, Japan. More than two million people were registered on this research panel. The sample size was calculated using a margin of error of 5%, a confidence level of 95%, a response distribution of 50%, and targeted population is 110 million, giving a minimum sample size of 1067 [13,14]. Therefore, the sample consisted of 1100 respondents, which was found to be suitable for similar European research [9].

The questionnaire sought the following information: (1) Sex; (2) Age; (3) Place of residence; (4) Whether having chronic condition (for example; hypertension, hyperlipidemia, diabetes, etc., not including for COVID-19 infection); (5) Vaccine willingness for COVID-19 vaccine (Would you be willing to be vaccinated when the COVID-19 vaccine is developed? Yes, Unsure, No); and (6) Reasons for previous answer (multiple answers).

Descriptive statistics were used to evaluate vaccine willingness by gender, age group, place of residence, and whether the respondent was presently ill. While indicating place of residence, prefectural levels were originally used; however, since these levels were too detailed to understand the tendency, we dichotomized the areas as the Central area (Kanto area: around the Tokyo metropolis and Kansai area: around Osaka metropolis) and Other areas. The reasons for wanting, not wanting, and hesitating to be vaccinated were evaluated by gender. For each statistical analysis, the chi-squared test was used to evaluate categorical variables. We also analyzed the characteristics of the main reason for vaccine hesitancy and not wanting to be vaccinated through logistic regression analysis, with gender, age, place of residence, and whether the respondent was presently ill as independent variables. The significance level was set at <0.05. JMP Pro 14.1.0 (SAS Institute Inc., Cary, NC, USA) was used for all the analyses.

The study was approved by the ethical committee of Kawasaki Medical School (Approval number: 5016-00). Implied consent was used rather than formal written consent to maintain the anonymity of participants. The participants clicked the “I agree” button before commencing the survey to indicate their consent.

## 3. Results

Of the 1100 participants, 584 were male (53.1%) and the average age was 44.8 years. A total of 645 (58.6%) were living in the central area, and 283 (25.7%) were presently ill (Table 1).

Overall, 723 (65.7%) of 1100 participants stated that they were willing to be vaccinated against COVID-19 if a vaccine was available. Another 242 (22.0%) of the participants stated that they were not sure and 135 (12.3%) stated that they did not want to get vaccinated. As shown in Table 2, we found considerable differences in the willingness to get vaccinated across genders, age groups, the prevalence of chronic condition, and places of residence.

A significantly higher proportion of men were willing to get vaccinated (68.0%, chi-squared, *p* = 0.014) than women (63.2%). The highest willingness to be vaccinated was found among the oldest age group (over 70 years, 77.2%, chi-squared, *p* = 0.002), while the greatest amount of uncertainly was found among 20- to 29-year-olds and 40- to 49-year-olds (28.0%). The proportion of 50- to 59-year-olds who were unwilling to get vaccinated was the largest with 19.0%. A significantly higher proportion of participants who had chronic diseases were willing to get vaccinated (78.4%, chi-squared, *p* < 0.001) than those who had not chronic disease (61.3%). Participants from rural areas were more willing to get vaccinated (70.3%, chi-squared, *p* = 0.026) than those from central areas (62.5%).

We asked participants who were willing to get vaccinated about their main reasons for this (Figure 1).

Almost all the participants said that they thought that vaccination was effective and a strong and preventable tool both for themselves (N = 625, 86.4%) and for the people around them (N = 437, 60.4%). However, approximately one-quarter of the participants (N = 154, 21.3%) thought that after vaccination, they would not need to continue engaging in preventive measures such as social distancing, using masks, and so on.

We asked participants who were unsure about being vaccinated about their main reasons for this (Figure 2). Nearly two-thirds said that they were concerned about the potential side effects of a vaccine, although this concern was more common among women than men. The same tendency was shown regarding their perception of the safety of the vaccination itself. Nearly one-fifth of the participants did not trust vaccine efficiency.

We found a similar trend among the most frequently mentioned reasons in terms of concerns of side effects among those who were not willing to get vaccinated (Figure 3).

Notable gender differences could also be observed among those participants who stated that they thought that COVID-19 was not dangerous to their health; the participants comprised more than twice as many men (N = 17, 12.6%) than women (N = 7, 5.2%).

Furthermore, logistic regression analyses revealed that age (OR: 0.991, 95% CI: 0.983–0.998), gender (male vs. female; OR: 1.496, 95% CI: 1.108–2.020), place of residence (Others vs. Central area; OR: 1.566, 95% CI: 1.156–2.121), and whether having chronic condition (One or more vs. None; OR: 1.935, 95% CI: 1.323–2.831) were significant factors associated with concerns about potential side effects of a COVID-19 vaccine (Table 3).

## 4. Discussion

The findings of our research revealed that 65.7% of the participants were willing to get vaccinated against COVID-19 if a vaccine was available. Upon comparing with previous research from European countries, France and Germany showed a very similar tendency (60–70%) to Japan, while the UK and Denmark had a very high willingness ratio (76–80%) [9]. Upon observing the composition ratio for willingness, uncertainty, and unwillingness to vaccinate, the tendency among Japanese people (the ratio in respective order: 65.7%, 22.0%, 12.3%) was more similar to the tendency exhibited in France (62%, 28%, 10%) [9]. The WHO SAGE Working Group classified the reasons and factors for vaccine hesitancy and refusal as follows: (1) contextual (due to historical, socio-cultural, environmental, institutional, economic, or political factors); (2) individual and group (personal beliefs and attitudes about prevention or previous experiences with vaccinations); and (3) vaccine/vaccination-specific (concerns about a new vaccine or formulation or about mode of administration or delivery) [11]. The issue pertaining to the human papilloma virus (HPV) vaccination for teenage girls is notorious among contextual reasons for vaccine refusal in Japan. The HPV vaccine against cervical cancer was expected to be such a powerful preventive measure that subsidies in local governments for the HPV vaccination programs commenced in 2010, and it became a vaccination with a public expenditure grant in April 2013 in Japan. However, repeated news reports regarding so-called adverse events such as chronic pain and motor impairment emerged soon afterwards, stoking public doubts about the vaccine’s safety. Consequently, the HPV vaccination rate among the younger generation in Japan has sharply decreased, with the percentage for females born after 2002 being 1% or less [15]. Although in some countries, such as the US, Greece, and Hong Kong, fear concerning the adverse effects of the HPV vaccination has recently increased and has become a significant reason for avoiding the vaccination [16,17,18], no other country has such a low rate. The over-announcement of adverse side effects may lead to an extremely low vaccination rate. Similar case was reported about influenza vaccination. Influenza vaccinations were previously included in routine immunization programs in Japan. However, skepticism regarding their efficacy increased in the 1980s and 1990s. This was partly attributed to studies that evaluated effectiveness based on the level of protection afforded against not just influenza, but “influenza-like illnesses”, which often include the common cold [19]. This design of these studies is understandable considering that no rapid diagnostic tests for influenza were available at that time. However, their findings may have led to an underestimation of the effectiveness of influenza vaccinations [20]. A similar logic applied to the general public, who had no measures to distinguish influenza from common colds, and often mistakenly underestimated effectiveness when they caught a cold after being vaccinated [19]. This skepticism led to an amendment to the Preventive Vaccination Law, which excluded influenza from routine immunization programs [20]. Although vaccination coverage has since improved, skepticism regarding efficacy still lingers today [21].

Our findings about the place of residence are also interesting. A previous study has suggested that adults living in urban areas had a stronger awareness of COVID-19 prevention than those in rural areas [22]. The results of our study contradict this finding. People living in rural areas were keener to be vaccinated than those in urban areas. The possible reasons for this result are the differences in population composition ratio and stigmatization of COVID-19 patients. The elderly population in rural areas is higher than in central urban areas [23]. The age group of over 70 years had the highest willingness to be vaccinated, thus, vaccine willingness in rural areas is also high. In addition, discrimination against COVID-19 patients in rural areas is stricter more blame is imposed on patients there than in urban areas [24].

Another previous Japanese study showed that 20% of the participants are reluctant to implement proper prevention measures. The characteristics of these groups are male, young (under 30 years old), unmarried, in lower-income households, possessing a drinking or smoking habit, and having a higher extraversion score [25]. Upon comparing and combining these results with our study results, a similar tendency can be noted. The younger generation and male population are more likely to refuse the vaccine than other groups.

Upon observing the reasons for vaccine refusal and being unsure about getting vaccinated, the main reason for these sentiments can be seen to be the fear of side effects. Logistic regression analysis revealed that the younger, female, unburdened by chronic disease, and central areas’ people groups felt more concern about side effects. Most vaccines have side effects, and because COVID-19 vaccines are quite new, the additional side effects other than the ones identified from the clinical trials are unknown. This fear is understandable and prevalent. It is, therefore, extremely important to ensure transparency in the information provided about the COVID-19 vaccines in terms of not only their efficacy but also side effects. The foundation of vaccination acceptance is public trust, namely, trust in vaccines and vaccine producers, in the healthcare profession, and the government [26].

This study has some limitations. First, this study was internet-based, hence, we could not eliminate the selectivity bias. Second, this study was a cross-sectional study, so no causality could be established. In addition, participants may have been affected by the available news about the virus at the time of the survey (September 2020). The questionnaire was designed to be simple and easy-to-answer, so we could not evaluate other socio-demographic factors such as income, educational level, and daily habits. Furthermore, the questionnaire items were based on the previous European survey, therefore some items might be not suitable for Japanese people.

Despite these limitations, our findings are quite unique and novel in that they are the first to report on COVID-19 vaccine willingness among the Japan population. Given the high vaccine hesitancy and refusal ratio in other countries, considerable policy efforts may be required to make the transition from making a vaccine available to adequate vaccination rates.

## Figures and Tables

**Figure 1 vaccines-09-00048-f001:**
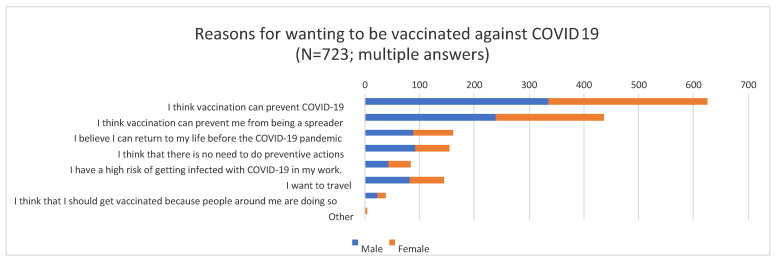
Participants’ reasons for wanting to get vaccinated against COVID-19.

**Figure 2 vaccines-09-00048-f002:**
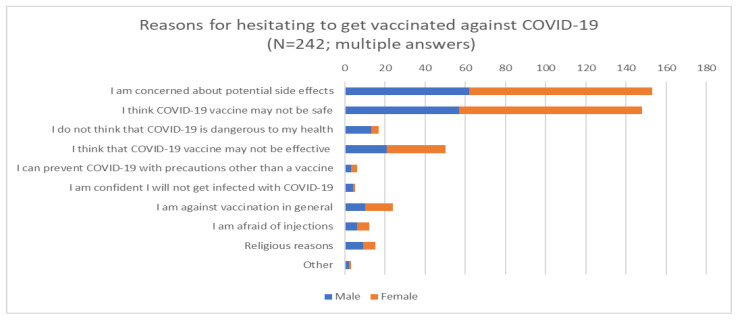
Reasons provided by participants who were unsure if they would like to be vaccinated against COVID-19.

**Figure 3 vaccines-09-00048-f003:**
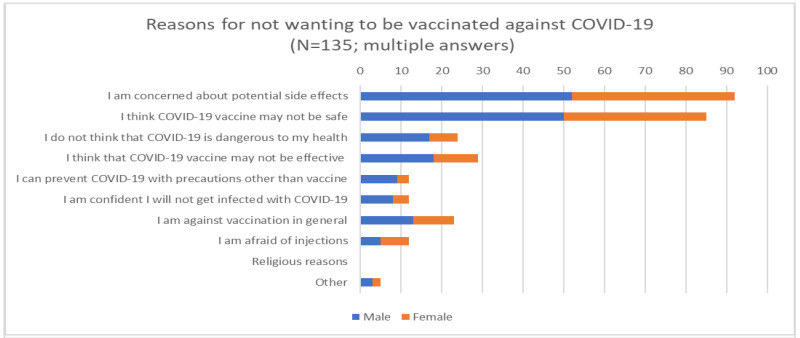
Reasons for not getting vaccinated against COVID-19.

**Table 1 vaccines-09-00048-t001:** Characteristics of respondents.

		N	%
Gender	Male	584	53.1
	Female	516	46.9
Age group	under 19	136	12.4
	20–29	211	19.2
	30–39	155	14.1
	40–49	161	14.6
	50–59	152	13.8
	60–69	118	10.7
	over 70	167	15.2
Chronic condition	None	817	74.3
	1+	283	25.7
Place of residence	Central	645	58.6
	Others	455	41.4

**Table 2 vaccines-09-00048-t002:** Willingness to be vaccinated against COVID-19 by characteristics.

		Yes (%)	Unsure (%)	No (%)	*p* *
Gender	Male	397 (68.0)	109 (18.6)	78 (13.4)	0.014
	Female	326 (63.2)	133 (25.8)	57 (11.0)	
Age group	under 19	94 (69.1)	28 (20.6)	14 (10.3)	0.002
	20–29	134 (63.5)	59 (28.0)	18 (8.5)	
	30–39	100 (64.5)	33 (21.3)	22 (14.2)	
	40–49	92 (57.1)	45 (28.0)	24 (14.9)	
	50–59	98 (64.5)	25 (16.5)	29 (19.0)	
	60–69	76 (64.4)	28 (23.7)	14 (11.9)	
	over 70	129 (77.2)	24 (14.4)	14 (8.4)	
(Repost)	<20	94 (69.1)	28 (20.6)	14 (10.3)	0.061
	20–59	424 (62.4)	162 (23.9)	93 (13.7)	
	>60	205 (71.9)	52 (18.3)	28 (9.8)	
Chronic condition	None	501 (61.3)	205 (25.1)	111 (13.6)	<0.001
	1 or more	222 (78.4)	37 (13.1)	24 (8.5)	
Place of residence	Central	403 (62.5)	155 (24.0)	87 (13.5)	0.026
	Others	320 (70.3)	87 (19.1)	48 (10.6)	

* Pearson’s chi-squared test.

**Table 3 vaccines-09-00048-t003:** Results of logistic regression analysis concerning potential side effects as dependent variables.

		AOR	95% CI	*p*
Gender	Male	1	-	
	Female	1.496	1.108–2.020	0.008
Age		0.991	0.983–0.998	0.025

Chronic condition	One or more	1	-	
	None	1.935	1.323–2.831	<0.001

Place of residence	Others	1	-	
	Central	1.935	1.156–2.121	0.003

AOR: Adjusted odds ratio. CI: Confidence interval. AICc: 1147.33, BIC: 1172.29, R^2^ = 0.025. Model was adjusted by gender, age, chronic condition, and place of residence.

## Data Availability

The data presented in this study are available on request from the corresponding author (T.Y.). The data are not publicly available due to privacy concerns.

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
