# Peer review of "Willingness to Receive COVID-19 Vaccination in Japan"

_vaccines, 2021, doi:10.3390/vaccines9010048_

Round 1
Reviewer 1 Report
Congratulate the authors for the work presented and their contribution in these difficult times of the global pandemic, shedding light on such important issues. In table 1 the values ​​of the columns are not clear, rewrite it. Table 3 Errors in editing overlapping letters.Author Response
Dear Reviewer 1,
Thank you very much for reviewing our manuscript and giving heartful comments.
We added the values of the columns in Table 1, and made new table as Table 3.
Additionally, We added some parts and re-wrote some points from first draft according to other reviewer's comments.
Reviewer 2 Report
Estimated Editors,
Estimated Authors,
thank you for the opportunity to review this very interesting paper on the Willingness to receive COVID-19 vaccination in Japan. Being on the eve of the mass vaccination campaigns, understanding and addressing main barriers towards vaccinations may help to improve immunization rates, accelerating the long expected return to the "old normality".
In my opinion, the present paper may be eventually published on the Vaccines journal, but some improvements are required.
1) Please include, in the discussion section, some comparisons with attitudes words vaccinations for highly communicable and severe infectious diseases in Japan. Authors have reported some insights from HPV vaccination campaigns, but obviously such comparison is not totally fitting the characteristics of SARS-CoV-2 pandemic.
2) in methods section it remains rather unclear how sample size was eventually calculated: Authors state that "the sample size was determined based on similar European research". The reference (10.1007/s10198-020-01208-6) does not include any sample size calculation, stating that "To shed more light on the issue of willingness to be vaccinated, we investigated people attitudes about vaccination against COVID-19 in an online survey among representative samples of the population (in terms of region, gender, age group and education) in seven European countries (N=7.662). The sample consisted of about 1.000 respondents per country, and an additional 500 from the highly affected region Lombardy, since we expected that results might differ from the rest of Italy". In other words, have you performed a preventive power analysis or, more simply, did you have followed WHO recommendations to include at least 1000 respondents from the target country? Please explain.
3) Barriers and facilitators towards COVID-19 vaccination are seemingly derived from the aforementioned article. This is interesting as allows a direct comparison with European data, but it risks to fail in addressing specific characteristics of the Japanese people.
4) No specific analysis of factors associated with a proactive vaccination status has been performed. It would be very interesting, for example, assessing whether the perceived risk of side effects are more extensively reported in certain population subgroup or not, possibly through a multivariate analysis.
As a minor remark, please be aware that table 1 lacks of its headings.
Author Response
Dear reviewer 2,
Thank you very much for carefully reading our manuscript. We appreciate you for your helpful suggestions. We have added some new findings according to your advices and have accordingly revised our manuscript. We have provided point-by-point responses to your comments.
Point 1: Please include, in the discussion section, some comparisons with attitudes words vaccinations for highly communicable and severe infectious diseases in Japan. Authors have reported some insights from HPV vaccination campaigns, but obviously such comparison is not totally fitting the characteristics of SARS-CoV-2 pandemic.
Response 1: Thank you for your suggestion. We added one episode about Japanese influenza vaccination hesitancy in the “Discussion” section (page 7-8, Line 204-217) according to your opinion.
Point 2: in methods section it remains rather unclear how sample size was eventually calculated: Authors state that "the sample size was determined based on similar European research". The reference (10.1007/s10198-020-01208-6) does not include any sample size calculation, stating that "To shed more light on the issue of willingness to be vaccinated, we investigated people attitudes about vaccination against COVID-19 in an online survey among representative samples of the population (in terms of region, gender, age group and education) in seven European countries (N=7.662). The sample consisted of about 1.000 respondents per country, and an additional 500 from the highly affected region Lombardy, since we expected that results might differ from the rest of Italy". In other words, have you performed a preventive power analysis or, more simply, did you have followed WHO recommendations to include at least 1000 respondents from the target country? Please explain.
Response 2: Thank you for your recommendation. We explained sample size calculation and added it in the “Materials and Methods” section (page 2, Line 56-59).
Point 3: Barriers and facilitators towards COVID-19 vaccination are seemingly derived from the aforementioned article. This is interesting as allows a direct comparison with European data, but it risks to fail in addressing specific characteristics of the Japanese people.
Response 3: Thank you for your opinion. We added limitations in the “Discussion” section (page 8, Line 249-251).
Point 4: No specific analysis of factors associated with a proactive vaccination status has been performed. It would be very interesting, for example, assessing whether the perceived risk of side effects are more extensively reported in certain population subgroup or not, possibly through a multivariate analysis.
Response 4: Thank you for your appropriate advice. We re-analyzed about logistic regression with concerning about potential side effects as dependent variables and items of gender, age, place of residence, and whether presently ill as independent variables. The additional analysis methods were described at the “Materials and Methods” section (page 2, Line 72-75) and the results were described “Results” section (page 6-7, Line 165-169, and Table 3). Additionally, our opinion based on the logistic regression result were described at the “Discussion” section (page 8, Line 235-237).
Point 5: As a minor remark, please be aware that table 1 lacks of its headings
Response 5: Thank you for your advice. We added items in the Table 1.
Reviewer 3 Report
Yoda T and Katsuyama H launched an internet-based questionaire survey about the willingness of taking COVID-19 vaccination among 1100 Japanese people. The results show about 65.7% of the responders were Yes, while 22.0% were Not sure and 12.3% were No, to take the vaccination. The underlying reasons have been analyzed. Basically, the survey is interesting and also helpful for local CDC's policy making. However, there are defects in the survey and analysis.
- Among the responders, there was a group of "presently ill". What kind of disease they suffered during the survey? If it was COVID-19, this group is definitely not the in scope of vaccination, thus should be excluded from the survey.
- Table 1 is not well prepared. No item in the first row? about the age group, the responders are better to group according the vaccination preffered age group, which is 20-59, thus the groups are <20, 20-59, and >60. About the "Presently ill" groups, does "1+" means more than one diseases they suffered?
- There is nothing new in the figures 1 and 2.
Author Response
Dear reviewer 3,
Thank you very much for reviewing our manuscript and offering valuable advice. We have provided point-by-point responses to your comments.
Point 1: Among the responders, there was a group of "presently ill". What kind of disease they suffered during the survey? If it was COVID-19, this group is definitely not the in scope of vaccination, thus should be excluded from the survey.
Response 1: Thank you for your suggestion. We added explain about presently ill (page 2. Line 62).
Point 2: Table 1 is not well prepared. No item in the first row? about the age group, the responders are better to group according the vaccination preffered age group, which is 20-59, thus the groups are <20, 20-59, and >60. About the "Presently ill" groups, does "1+" means more than one diseases they suffered?
Response 2: We apologized for not well prepared about Table 1. We added items in the Table 1. And according to your opinion, some items were changed and added in the Table 1 and Table 2.
Point 3: There is nothing new in the figures 1 and 2
Response 3: Thank you for your opinion. As your opinion, figures 1 and 2 were not unusual in the other countries without Japan, but such kind of Japanese research results were seldom found in English paper. Therefore we described it.
Round 2
Reviewer 3 Report
The mansucript has been improved after the revising.
Author Response
thank you